# Management of Hardware Vulnerabilities in the Life Cycle Stages of Microprocessors and Computers

**Ignat Bychkov [1], Irina Mikhailova [2], Pavel Korenev [2], Vitaliy Pikov [3] and Anatoly Ryapukhin [3,*]**

[1] Moscow Institute of Physics and Technology, Institutskiy Lane 9, 141701 Dolgoprudny, Russia; ignatbychkov@mail.ru

[2] Joint Stock Company "The Institute of Electronic Control Computers named after I.S. Bruk", Vavilov Street 24, 119334 Moscow, Russia; irina.mikhailova.ira@mail.ru (I.M.); pavel.v.korenev@mail.ru (P.K.)

[3] Moscow Aviation Institute, Volokolamskoe Highway 4, 125993 Moscow, Russia; vitaly_pikov@mail.ru

[*] Correspondence: ryapukhin.a.v@mail.ru

**Abstract:** This article discusses the topical issues of managing information security vulnerabilities in the life cycle stages of processors and computer equipment. An analysis of the experience of identifying vulnerabilities in the course of the joint design of the processor, computing module and computing complex was carried out. A number of actions have been developed and presented to ensure the control of hardware vulnerabilities in the development stage. The use of the binary translation technology of the Elbrus platform is proposed to prevent the execution of malicious software. A method has been developed to eliminate vulnerabilities in computer equipment for automated systems used for various purposes by using the Lintel binary translation system component. An experiment is described, the purpose of which was to successfully exploit the Meltdown vulnerability on a computer with an Elbrus processor. The experiment showed that, due to the peculiarities of the microarchitecture of Elbrus processors, the exploitation of Meltdown-type vulnerabilities is impossible.

**Keywords:** information security; vulnerabilities; microprocessor; computer technology life cycle; integrated circuit design; binary translation; Meltdown

## 1. Introduction

The activity of humankind in the field of information protection began a long ago. After all, people know how to keep their secrets. But, as they say, the "devil is in the details". The news includes reports about successfully carried out hacker attacks or leaks of the client bases of companies with big names. Yes, the world is still far from perfect in terms of information security. Another confirmation of this is the constant change in regulatory legal acts: there are many legislatively significant documents regulating the activities of the Russian Federation in this area.

The normative basis for information security should be relevant, because it is the basis of the work of information security specialists. In accordance with the regulatory legal framework, the means of protecting information are being changed and improved. The whole set of software and hardware information security tools is constantly "working" for the benefit of the security of confidential and protected information in Russia.

A hacker introduces a protected, from an informational point of view, object through "holes" in the security system: vulnerabilities. There are a huge number of vulnerabilities in all programs, software and hardware solutions that underlie any information system. In view of the fact that programmers write programs and engineers create computers, it can be taken as an axiom that there will always be errors and flaws, which means that computers will always have vulnerabilities. It turns out that software will always be insecure. Even if the program code is checked by several specialists, there will be a researcher who is able to

find an error and a way to use it to attack a computer, server, service, application or entire computer network.

Separately, it is worth highlighting the experience of Russian companies in creating secure software. For example, in Russia, this issue has long been dealt with by the Federal State Budgetary Institution of Science "Institute of System Programming named after. V. P. Ivannikov" of the Russian Academy of Sciences and the company JSC NPO "Echelon" under the supervision of the Federal Service for Technical and Export Control of Russia, represented by highly qualified specialists of the technical committee for standardization "Information Protection" (TC 362).

Not so long ago, in 2016, the State Standard GOST R 56939-20XX "Information Protection. Development of secure software. General requirements" was presented. After a heated and fruitful discussion, this standard was approved. Currently, GOST R 56939-2016 is the current standard of the Russian Federation in terms of creating secure software. In a speech by the Deputy Director of the Federal Service for Technical and Export Control of Russia, Vitaly Sergeevich Lyutikov, in February 2023 at the conference "Topical issues of information security", "TB Forum 2023", information was provided indicating that work in this direction is ongoing, and a number of guiding documents and state standards will soon appear (after being read by specialists in various fields from a number of organizations) and be put into effect once approved [1]. Documents that have been prepared include a guide to software development security assessment, a draft methodology for monitoring (analyzing) the security of information systems and a methodology for managing vulnerabilities in an organization, guidelines for conducting static and dynamic software analysis, trusted C/C++ language compilers, a methodology for developing trusted systems and ensuring constructive information security, and software security management using borrowed and contracted components.

At the moment, the Russian Federation is actively pursuing the tasks of ensuring breakthroughs in scientific, technological and socio-economic development. For example, the Decree of the President of the Russian Federation of 7 May 2018 No. 204, "On National Goals and Strategic Objectives for the Development of the Russian Federation for the period up to 2024", defines the priority development goal as the accelerated introduction of digital technologies and platform solutions in the field of public administration, and one of the most important tasks is the digital transformation of public administration.

Many federal executive authorities have adopted general technical requirements for Russian hardware and software platforms that regulate the development of high-performance processors that underlie modern computing [2]. These general technical requirements have been developed to ensure the required level of technological independence and information security and largely govern the development of special-purpose computing systems. The introduction of these requirements, in combination with other regulations (federal laws (FLs) and government decrees (GDs)) of the Russian Federation, is aimed at the development of Russian microelectronics (Figure 1).

The task of minimizing the number of vulnerabilities in operated information systems is of concern not only to Russian professionals in the field of information security. Large foreign companies such as "Microsoft", "Cisco", "IBM", "Hewlett Packard Enterprise" and "Google" have their own visions of organizing the development of secure software. A wealth of experience has been accumulated in creating small programs and large-scale global projects for the implementation of information technologies, taking into account the requirements for ensuring information security.

Safe hardware and software products can be such products, in which the number of vulnerabilities is minimized due to the methods and corresponding tools used in all stages of their life cycle. It is the availability of special software tools, such as program code analyzers, security scanners and integrated security analysis systems, that makes it possible to minimize the appearance of software vulnerabilities in the creation stage or identify them during operation.

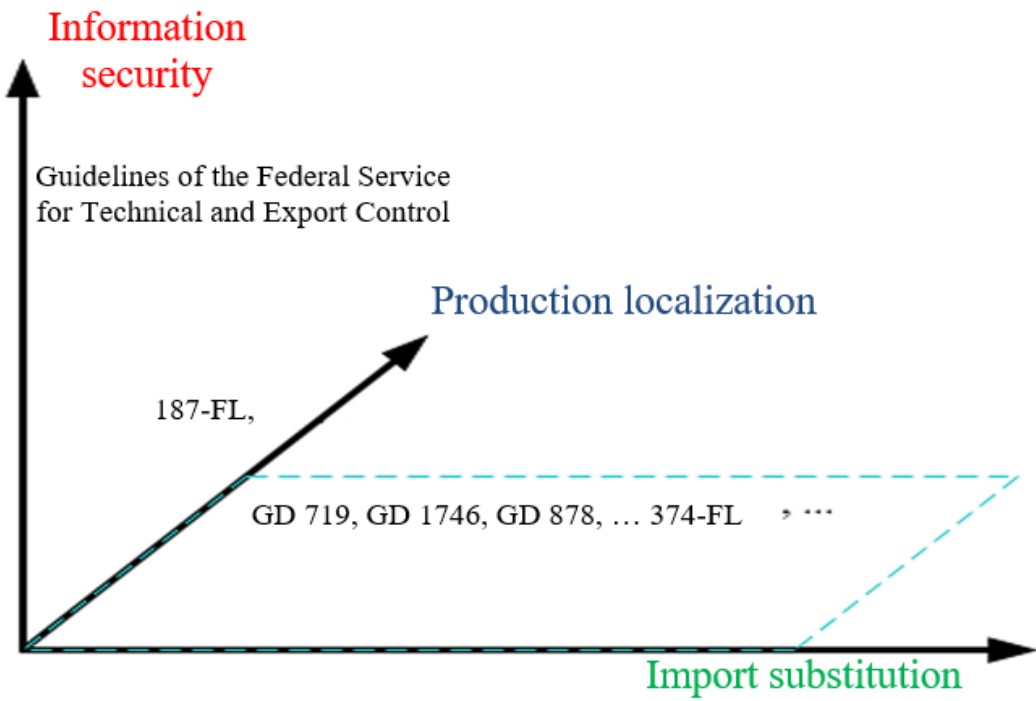

**Figure 1.** Regulation of the development of electronics and information technology.

Static Application Security Testing (SAST) and Dynamic Application Security Testing (DAST) methods have shown their effectiveness in identifying vulnerabilities in program source codes. SAST is a testing technique from the point of view of a programmer and developer, and DAST allows the testing of software by exposing it to attacks and non-standard actions by users.

There are three key problems associated with the search for vulnerabilities in hardware and software solutions:

- The lack of generally accepted standards and methods for finding vulnerabilities during the initial stages of development. Many small businesses and start-ups are unable to use costly enterprise solutions and often use their own methods, which are not always effective.
- The lack of a sufficient number of qualified specialists to detect vulnerabilities in hardware and software solutions. Many small businesses and start-ups are unable to bring in experienced professionals to solve problems associated with the discovery of vulnerabilities and are forced to solve these problems themselves.
- A lack of sufficient awareness of existing vulnerabilities and how to detect them among small businesses and start-ups. Many developers of hardware and software solutions do not have sufficient experience with information security and do not know what vulnerabilities can arise during the development process.

"We are exploring vulnerabilities as a source of information security threats. Currently, in open sources, there is no single classification of vulnerabilities, while there are many attempts to restore order in this area on the Internet. As a rule, one of two well-known classifications of vulnerabilities is implied:

First classification:

- vulnerabilities that arose at the design stage;
- vulnerabilities that arose during implementation;
- vulnerabilities allowed in the configuration.

Second classification:

- incorrect processing (checking) of the input data of the system;
- weak authentication mechanisms;

- insufficient quality data authentication;
- software configuration errors;
- incorrect use of cryptography methods;
- mistakes made in the process of managing accounts" [3].

The requirements of GOST R 56545-2015, "Information security. Vulnerabilities of information systems. Rules for describing vulnerabilities", establish the classification of vulnerabilities, the rules for describing vulnerabilities, the content and the procedure for performing work to identify and assess the vulnerabilities of information systems. The standard adopts rules for describing vulnerabilities that can be used by information security specialists when creating and maintaining a database of information system vulnerabilities, developing information security control (analysis), developing models of information security threats and designing information security systems, carrying out work to identify, analyze and eliminate vulnerabilities. "The standard does not apply to information system vulnerabilities associated with information leakage through technical channels, including vulnerabilities in electronic components of technical (hardware and firmware) information systems" [4].

Numerous databases are openly available on the Internet that contain information about hundreds of thousands of vulnerabilities found in software. The vulnerability databases recognized as the most authoritative among information security specialists include the National Vulnerability Database (NVD), Common Vulnerabilities and Exposures (CVE) and Open Vulnerability and Assessment Language (OVAL).

We study the concept of "vulnerability" and the terms associated with this phenomenon. Here, we provide definitions from international standards in the field of information security, as well as from the official website of the Federal Service for Technical and Export Control of Russia. According to ISO/IEC 27000:2014, "A vulnerability is a weakness in an asset or management, the exploitation of which will lead to the realization of one or more threats" [5].

In accordance with GOST R 56546-2015, "Information security. Vulnerabilities of information systems. Classification of vulnerabilities of information systems", the "classification of vulnerabilities of information systems, based on the area of origin of vulnerabilities, types of deficiencies in information systems and places of occurrence (manifestation) of vulnerabilities in information systems" is adopted [6].

In accordance with the standard, "the following classification features are used as the basis for the classification of information system vulnerabilities:

- area of origin of the vulnerability;
- types of deficiencies in information systems;
- place of occurrence (manifestation) of vulnerability of information systems".

"The following are considered as vulnerable components of an information system: system-wide (general), applied, special software, hardware, network (communication, telecommunication) equipment and information security tools" [6].

In accordance with clause 5.1 of the standard, "information system vulnerabilities by area of origin are divided into the following classes:

- code vulnerabilities;
- configuration vulnerabilities;
- architecture vulnerabilities;
- organizational vulnerabilities;
- multifactorial vulnerabilities".

In accordance with the requirements of clause 5.2, "information system vulnerabilities by types of information system deficiencies are divided into:

- deficiencies associated with incorrect configuration of software parameters;
- shortcomings associated with the incompleteness of the verification of input (input) data;
- shortcomings associated with the ability to trace the path of access to directories;

- disadvantages associated with the ability to follow links;
- shortcomings associated with the possibility of introducing operating system commands;
- disadvantages associated with cross-site scripting (script execution);
- disadvantages associated with the introduction of interpreted statements of programming languages or markup;
- disadvantages associated with the introduction of arbitrary code;
- disadvantages associated with memory buffer overflow;
- disadvantages associated with an uncontrolled format string;
- computational deficiencies;
- deficiencies leading to leakage/disclosure of restricted information;
- shortcomings associated with the management of powers (credentials);
- deficiencies related to the management of permissions, privileges and access;
- weaknesses associated with authentication;
- deficiencies associated with cryptographic transformations (encryption deficiencies);
- disadvantages associated with cross-site request spoofing;
- deficiencies leading to a "race condition";
- deficiencies related to resource management;
- other types of shortcomings" [6].

In accordance with clause 5.3 of the standard, "information system vulnerabilities by the place of occurrence (manifestation) are divided into:

- vulnerabilities in system-wide (common) software;
- vulnerabilities in application software;
- vulnerabilities in special software;
- vulnerabilities in technical means;
- vulnerabilities in portable hardware;
- vulnerabilities in network (communications, telecommunications) equipment;
- vulnerabilities in information security tools" [6].

This study focuses only on vulnerabilities classified by:

- Area of origin: architecture vulnerabilities;
- Types of information system deficiencies: deficiencies associated with computing and deficiencies leading to leakage/disclosure of restricted information;
- Place of occurrence (manifestation):
- Technical vulnerabilities.

Based on the general technical requirements for hardware and software platforms in the life cycle stages of processors, it is necessary to keep an effective account of vulnerabilities in the hardware being created. At the place of localization, vulnerabilities in the processor may occur. Examples of actual vulnerabilities localized in the processor are shown in Table 1. It is important to note that manufacturers pay special attention to the issue of accounting for vulnerabilities in processors used abroad. For example, at Intel, a large division has been allocated to account for and check vulnerabilities in a large range of products [7].

The processor vulnerabilities under consideration lead to unforeseen functional properties of processors and are shortcomings in computing technology based on it: undeclared capabilities that create potential or actual conditions for the implementation of information security threats. The specificity of eliminating vulnerabilities in the hardware of computer equipment identified after the release of the processor is the high cost of a new iteration of a semiconductor chip. At the same time, the elimination of vulnerabilities is possible only with the internal development of the architecture for processor cores and the memory subsystem. At the moment, the country only carries out its own development of the architectures of processor cores for Elbrus and "RISC V" platforms. As a rule, the elimination of vulnerabilities in the hardware of computer equipment by changing the general or system-wide software leads to a significant decrease in the performance of computer systems as a whole.

**Table 1.** Examples of popular vulnerabilities in Intel processors.

| Vulnerability/Violation | Processor, Year of Manufacture | Brief Description of the Vulnerability |
|---|---|---|
| Dual Sigma/accessibility | Intel 80386/ 1985–1986 | Intel 80386 processors (even before 386DX and 386SX) could hang when executing 32-bit code. The company could not find faulty processors during production. |
| Pentium FDIV/integrity | Pentium 60/66 MHz/ 1993–1996 | There was an error while dividing floating-point numbers using the FDIV command, and the result could be incorrect. |
| God Mode/integrity, accessibility, confidentiality | All Intel processors 1995–2010 | System Management Mode (SMM) was used to debug the processor, which suspended the execution of any third-party code and ran a special program stored in a protected memory area. All access rights were obtained. |
| Spectre CVE-2017–5753 and CVE-2017–5715/confidentiality | All Intel processors 1995–2017 | The ability to analyze isolated user data in programs was gained using speculative calculations. |
| Meltdown CVE-2017–5754/ confidentiality | All Intel processors 1995–2017 | There was an ability to bypass memory isolation measures and gain read access to the operating system's memory, which led to the ability to analyze user data, including customers of cloud infrastructures. |
| Foreshadow/L1 Terminal Fault (CVE-2018-3615, CVE-2018–3620, CVE-2018–3646)/confidentiality | Intel Core and Xeon Processor E3 processors 2015–2017 | There was a composite vulnerability in Software Guard Extensions (SGX) data protection technology, consisting of three components: vulnerability in Intel SGX, operating system kernel and SMM, virtualization software and Virtual Machine Monitors. |

On the current topic of cybersecurity, areas such as the detection of new threats to information security, vulnerabilities in systems and programs, attacks, intrusion detection in systems and software, network traffic anomalies and the development of secure software have been explored by Russian and foreign authors, such as O.V. Kazarin, V.V. Lipaev, A.V. Barabanov, A.S. Markov, A.I. Kachalin, N.A. Gorbunov, V.L. Tsirlov, I.V. Kotenko, A.V. Lukatsky, I.V. Sharabyrov, E.S. Abramov, D.A. Andreev, M.L. Evgenievich, D. Yu. Gamayunov, A.N. Nazarov, Kh.A. Foelevich, A.A. Vladimirov, V.A. Galatenko, S.V. Gordeychik, S.A. Ermakova, P.D. Zegzhda, Constantinos Kolias, Muhamad E.A., Kwangjo Kim, M. Usha, et al. [8–19]. An analysis of these sources shows that well-known works do not consider the features of the development of processors, common software and general system software, nor do they take into account vulnerabilities at the stages of the life cycle of processors and computer equipment. It is not fundamental research but rather developments aimed at practical applications that will increase the level of information security of various organizations and enterprises.

The fuzzing testing method used when testing software code does not allow for the complete enumeration of parameters and settings due to high time and computational complexity; therefore, it is advisable to conduct scientific research on promising methods to improve the efficiency and speed of finding vulnerabilities. It should also be noted that well-known Russian software implementations of well-known ways to search for vulnerabilities do not always take into account the factors of import substitution and sanctions by foreign states or the transition to Russian high-tech computing platforms. It is necessary to develop new approaches to finding vulnerabilities.

## 2. Materials and Methods

We analyze the features of the development of processors, general software and general system software. Experience in the joint design of a processor, computing module and computing complex has shown its effectiveness in solving the problems of minimizing and controlling errors in hardware during its creation. The process of implementing a computing complex can be visualized in the form of a typical timeline in a Gantt chart, as shown in Figure 2. Errors, in most cases, lead to vulnerabilities in the computing complex. In the process of designing performed in this way, a multicriteria optimization problem is solved, the parameters of which are design solutions using available technologies for the periphery of the crystal, the chip package and the computing module. It is also important to note that the parallel organization of a design and the pre-manufacturing of a product make it possible to reduce costs, identify potential vulnerabilities and significantly increase the competitiveness of products [20,21].

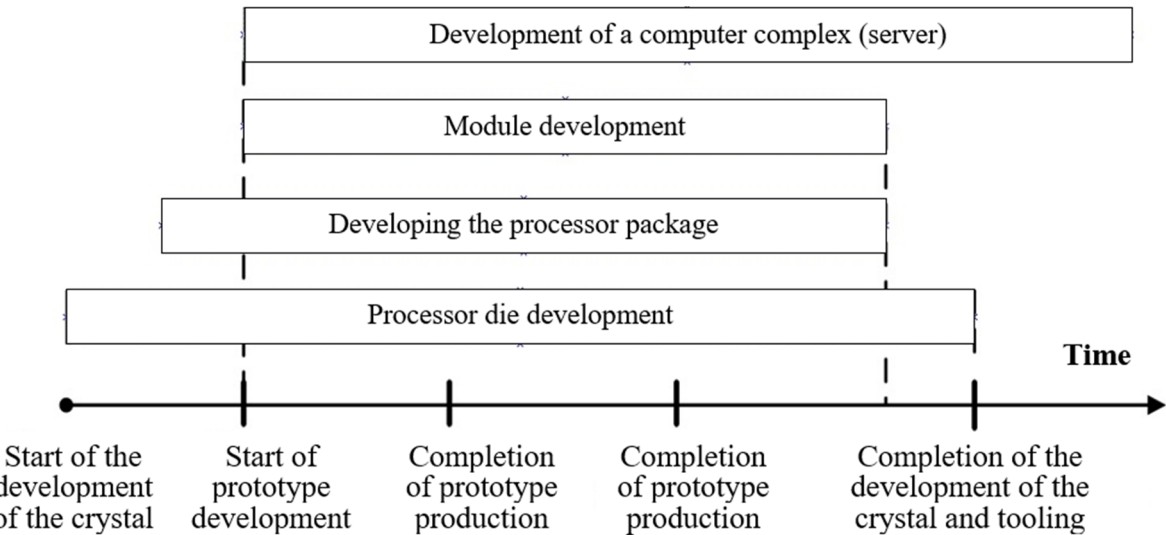

**Figure 2.** Gantt chart for the implementation of the computing complex.

The Elbrus processor design route includes the following steps:
- Development of logical circuits for complex digital blocks (CDBs), such as main processor cores or controllers, or their acquisition (licensing) in the case of graphics cores, memory controllers, input/output, etc.;
- CDB logic verification;
- Integration of CDBs into a single logic circuit;
- Verification of a single logical Very Large Scale Integrated Circuit;
- Development of a crystal topology based on a logic circuit;
- Transfer of the project to the factory to create photomasks for the manufacture of a semiconductor crystal;
- Development of a crystal case and transfer of the project to the factory for assembling the microcircuit;
- Testing and sorting of semiconductor crystals and microchips of processors.

For Elbrus processors, an additional development element is the creation of appropriate development tools, such as:
- Auxiliary design tools;
- High-level language compilers, etc.

According to the general technical requirements for a single type or range of Russian hardware and software platforms, all the main components of the processors are developed in-house. These include the central core, cache memory of all levels and a number of channel controllers for interprocessor and peripheral exchange.

As a part of the development of the processor, an in-house compiler is being created for the popular high-level languages C, C++ and Fortran. For the processor, the compiler plays a key role, especially for processors with Elbrus architecture. The compiler provides for the use of parallelism during the operation of the processor. The in-house compiler allows, in many cases, processor errors to be bypassed or effectively leveled out.

Special bootloader software, as an analog of BIOS made by the compiler, supports various models of computing modules on a given processor and has the functionality of transferring control to the trusted boot module to control the boot process of the operating system.

For processors, a proprietary operating system is being developed based on open-source codes: Linux kernel and many programs from popular distributions are used in total of more than 8000 packages. Based on the adapted Linux kernels, the practice of creating other operating systems has been worked out: Neutrino, Alt8 SP, Astra Linux SE and Synthesis.

We consider the process of the verification of processors during development. The development of Elbrus processors is carried out according to the route developed for these purposes. The Elbrus processor development route complies with the requirements of Russian standards and is intended for implementation by specialists:

- Developers of complex-functional processors;
- Verification specialists.

The document regulates the procedures for the provision and control of technologies for the development and/or production of complex-functional semiconductor crystals for products that implement information technologies in a secure design. The application of the route is carried out while taking into account the work of those responsible for the areas of hardware and software, who manage information security during the stages of the processor life cycle.

The main requirement for hardware verification in the product development process is the use of the necessary set of models and tests shown in Figure 3. These models and tests may vary slightly from project to project, but in general, the structure is preserved.

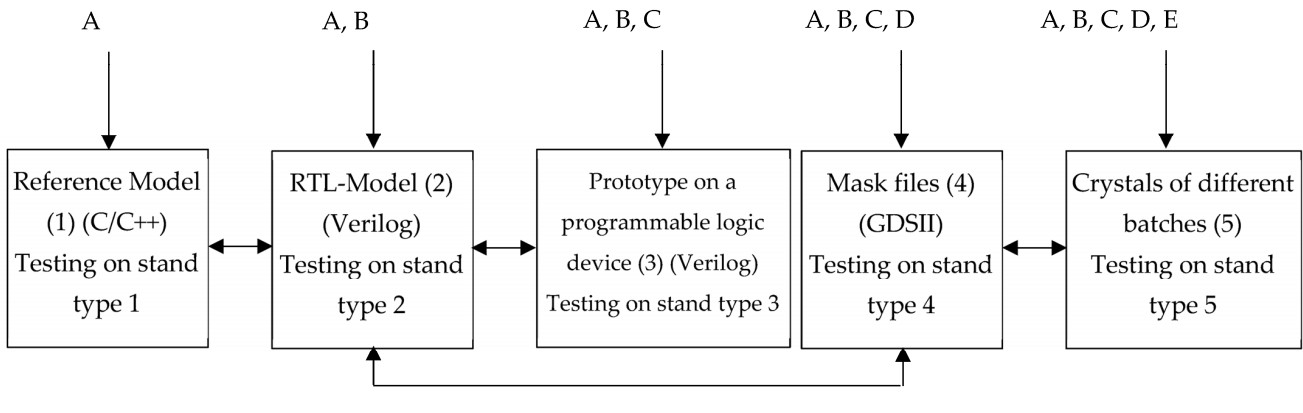

A – Architecture validation tests (AVS)

B – Stand-alone and directional device tests

C – Tests via Joint Test Action Group (JTAG) using built-in analyzers

D – Analysis/calculations of time characteristics, power, voltages, etc.

E – Tests and analysis of data from built-in diagnostic equipment (temperature sensors, voltages in the crystal and consumption currents of the stand, etc.)

**Figure 3.** Many models and tests for verification.

Information security management during the creation and production of complex-functional chips of processors and controllers is organized and carried out in accordance with the provisions of GOST R ISO/IEC 27001-2021 [22]. Such information security management is organized and carried out in order to:

- Prevent the leakage of confidential information regarding design decisions adopted for implementation in microchips, as well as prevent the leakage of information regarding the results of their testing, verification and confirmation of compliance;
- Minimize the risks of functional properties (qualities) not provided for by the terms of reference in the created microcircuits of processors and controllers being intentionally introduced into design solutions;
- Minimize the risks of missing the presence of functional properties not provided for by the terms of reference in the created microcircuits of processors and controllers in design solutions during testing, verification and confirmation of compliance;
- Organize the development of microchips of processors and controllers in specialized computer-aided design systems, which makes it possible to ensure minimal residual risks of the deliberate introduction of functional properties not provided for by the technical specifications into design solutions.

When developing and applying these models and tests, the following requirements must be met:

- All types of models and stands for testing are created by different departments of the enterprise;
- The model and its tests are developed by different employees of the enterprise;
- The determination of the set of tests takes into account the completeness of testing and the implementation of test plans;
- No stands or tests are transferred to other enterprises.

The basis of verification and testing is AVS, which is updated with each chip development project. This architecture validation database is restricted, contains information about the original architecture and is never shared with other enterprises.

During the stage of the physical design, the verification of the conformity of the developed circuitry of the processor (circuit diagram) and the initial description of the crystal (logic equivalence checking) are carried out using the verification tools of two independent suppliers. It is desirable to check the conformity of the developed topology and the circuit diagram of the crystal (layout versus schematic) using the verification tools of two independent suppliers.

In the process of checking the manufactured batches of microcircuits, original tests (type E in Figure 3) are used, which reveal the parameters of reliable operation through comprehensive analysis and the use of devices built into the semiconductor chip. The operability check is carried out in the limiting operating modes: by frequency, voltage, temperature, load tests, etc. At the same time, the operating ranges are necessarily monitored and compared with the results obtained on a type 4 bench according to Figure 3 [20].

Errors identified during the verification process are interpreted as vulnerabilities by the person responsible for the hardware and are documented with the definition of a potential threat to information security (violations of confidentiality, integrity, information availability). In this case, the percentages of errors by type of threat are indicated, as is outlined in Figure 4.

A description of all errors that have not been eliminated in batches of microcircuits intended for the manufacture of computer equipment is drawn up in a separate document agreed upon by those responsible for ensuring the information security of hardware and software. This document is approved by the chief designer of the microcircuit. It is important to note that, without fail, access to it is limited in the regime of a trade secret of the enterprise.

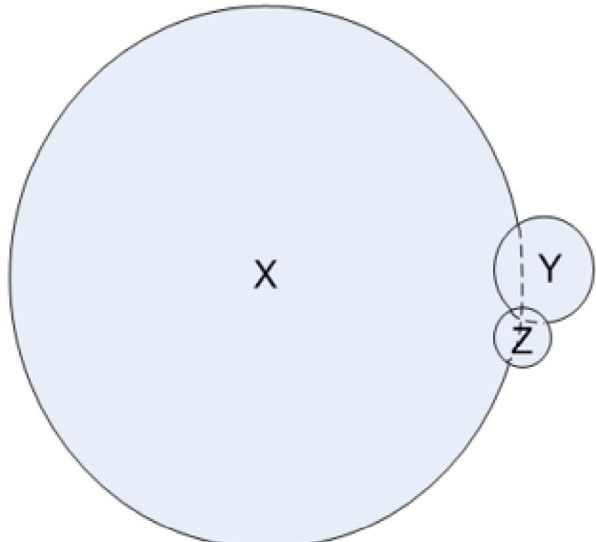

X – errors leading to a violation of the availability of information
Y – errors leading to a violation of the integrity of information
Z – errors leading to a violation of the confidentiality of information

**Figure 4.** Example of representing the ratio of errors by type of threat.

The duty of officials who manage information security at all life cycle stages of processors is to bring documented errors to the chief designers of computer equipment, with their subsequent consideration when developing threat models and violator models in created or upgraded products: automated systems for various purposes or control systems. The impact of documented hardware errors (vulnerabilities) is taken into account when creating information protection systems for products in accordance with the regulatory legal documents of regulators in the field of information protection and also when subsequently accepted for supply.

Also, to solve the problem of detecting vulnerabilities in hardware and software solutions, various methods and tools can be used:

- Manual testing. This method consists of manually checking hardware and software solutions for possible vulnerabilities, using experience and knowledge in the field of their security. Manual scanning can be used to detect more complex vulnerabilities that can be missed by an automatic scanner. This method has a number of disadvantages, such as high labor intensity, a long check time and limited detection of new vulnerabilities.

- Using automatic vulnerability scanners. These are software tools that automatically scan a web application for vulnerabilities and report the results of the scan. There are a lot of such tools, both commercial and free (Sn1per, Wapiti3, Nikto, OWASP ZAP, Sqlmap, etc.). However, despite its high speed and efficiency, this technique also has its drawbacks. Automated scanners can produce false positives and false negatives and fail to detect new and unknown vulnerabilities.

- Use of version control systems. These are designed to track changes in the application code and control application versions. Git, SVN, Mercurial, etc., can be distinguished among such systems. The use of version control systems makes it possible to detect changes associated with the addition of new vulnerabilities to the program code of hardware and software solutions.

- Use of artificial intelligence and machine learning. This method is one of the most promising methods for detecting vulnerabilities and consists of creating a learning model based on a large amount of data and training it to recognize types of vulner-

abilities in hardware and software solutions. This allows the detection of new and unknown vulnerabilities, as well as a reduction in the number of false positives.

There are also a large number of tools for detecting vulnerabilities in hardware and software solutions, which can be divided into several categories depending on how they analyze the application:

- Vulnerability scanners are programs that automatically scan web applications for vulnerabilities using predefined attack scenarios and vulnerability knowledge bases. Examples of such scanners are Acunetix, Nessus, Burp Suite, etc.
- Static analysis tools are programs that analyze the source code of web applications for vulnerabilities. This type of tool can find vulnerabilities that cannot be found by vulnerability scanners. Examples of such tools are Veracode, Checkmarx, SonarQube, etc.
- Dynamic analysis tools are programs that analyze the behavior of a web application while it is running. They allow the identification of vulnerabilities associated with the incorrect processing of user input, as well as the detection of vulnerabilities associated with incorrect server or database configuration. Examples of such tools are AppScan, WebInspect, Netsparker, etc.

In addition, there are many open-source research projects, such as OWASP ZAP and W3af, that allow the real-time vulnerability testing of web applications and the customization of attack scenarios for a specific application.

We consider the binary translation technology of the Elbrus platform. Through the binary translation technology of the Elbrus platform, the practice of executing various programs on it for foreign Intel or Advanced Micro Devices (AMD) processors is widespread. In particular, the application layer's own binary translation facility is used to run software on the use of secure magnetic storage media. Another application of system-level binary translation is to run binary operating systems in the popular $\times$86-64 instruction set, such as the Microsoft Windows family of operating systems. Converting non-target instruction set codes with optimizations can prevent malicious software from being executed. There are vulnerabilities inherent only in some hardware and software solutions. By a hardware–software solution, we mean the central processor used in the computer technology and the totality of the software used. In the general case, a vulnerability will be a violation of the state of information security of the system when the conditions for its implementation are met.

$$V(CVE) = \{Proc, Prog, Vector\}, \tag{1}$$

where *Proc* is the processor for execution, *Prog* is the set of programs for execution, and *Vector* is the attack vector resulting from program execution and the direction of actions for its application.

When using a binary translator, in many cases, there is the following inequality:

$$V(CVE) = \{Proc, Prog, Vector\} \neq \{Proc_N, Prog_N, Vector\}, \tag{2}$$

where $Proc_N$ is the used central processing unit, $Prog_N$ is the set of software used, *Vector* is the attack vector, which is the conditions for executing malicious code by exploiting a vulnerability, $Proc_N$ is the central processor for executing the converted (by a binary translator) code, and $Prog_N$ is the program code for execution on the target hardware–software solution (*native*) for which the search is performed for the exclusion of vulnerabilities.

Also, inequality (2) can be explained as follows: a vulnerability found on one hardware–software solution will not in all cases be reproduced on a hardware–software solution using a binary translator. Or even a more stringent statement applies: a vulnerability found on one hardware–software solution will not be reproduced on a hardware–software solution using a binary translator.

Therefore, an experiment should be conducted to test the method for eliminating vulnerabilities using the architectural features of the Russian Elbrus computing platform.

Previously, according to information from open sources, the possibility of eliminating vulnerabilities using a new method, which consists of applying the architectural features of the Russian Elbrus computing platform and binary translation technology, was not considered. This is the novelty of this study.

The following assumption was tested during the experiment: to recreate a well-known vulnerability, for example, Meltdown, using an implementation error of speculative command execution [23,24] in any of the operating systems of Microsoft Windows, Linux or macOS family on the Elbrus platform using a Lintel binary translator system, is not possible.

The Meltdown (rogue data cache load) vulnerability has been assigned the international code CVE-2017-5754. The description in the official source is as follows: systems with microprocessors utilizing speculative execution and indirect branch prediction may allow unauthorized disclosure of information to an attacker with local user access via a side-channel analysis of the data cache [25].

## 3. Results

We examined the process of providing control for the presence of hardware vulnerabilities.

At the moment, the joint-stock company "Moscow Center of SPARC Technologies" (MCST) is developing new general-purpose processors with a full production cycle in the Russian Federation. The system of the bit compilation or binary translation of new processors is constantly being improved.

The MCST website provides a description of the tool for launching operating systems in ×86 machine codes on computers with Elbrus architecture: the binary translator of the Lintel system (TVGI.00509-01).

"The binary translation system component, known as Lintel, allows you running an operating system in ×86 or ×86-64 machine codes, such as Microsoft Windows or Red Hat Enterprise Linux on Elbrus architecture computer without recompiling from source. The broadcast takes place in real time, "on the fly", with adaptive multi-pass optimization, which, in combination with the broadcast support hardware embedded in Elbrus architecture and providing low overhead and gives high speed of guest systems. Unlike the application translator, the system-level translator creates the most complete resemblance of a simulated ×86 computer" [26].

To confirm the main assumption of the study, an experiment was conducted, the purpose of which was to successfully exploit the Meltdown vulnerability on a computer with an Elbrus microprocessor. The repository with the program code for demonstrating the Meltdown vulnerability is located on the largest web service for hosting IT projects and their joint development (GitHub) [23].

The repository contains several applications that demonstrate the "Meltdown" vulnerability. For technical information about the vulnerability, see [27].

At the very beginning of the experiment, it was suspected that, given the microarchitecture features of Elbrus processors, the program code of the demo would not work.

The plan of the experiment, indicating the name of the demo and the description and interpretation of the results, is indicated in Table 2.

**Table 2.** Plan of the experiment and interpretation of the results.

| No. | Action | Description of Demo, Interpretation of Result |
|---|---|---|
| 1 | Installation on a personal computer with Elbrus-8C2 processor of a binary translator of Lintel system (TVGI.00509-01) and copy of Ubuntu operating system | The system is prepared for demo tests. |
| 2 | Download the source texts of the demos from the source and unzip them into a separate folder, run the make command | Executable files are obtained for testing. |

**Table 2.** *Cont.*

| No. | Action | Description of Demo, Interpretation of Result |
|---|---|---|
| 3 | Demo 1 | The Meltdown vulnerability is exploited to read available addresses from its own address space without violating any isolation mechanisms. |
| 4 | Demo 2 | The Meltdown vulnerability is exploited to leak (secret) direct physical map randomization. |
| 5 | Demo 3 | This demo tests how reliably physical memory can be read. This demo requires either the direct physical map offset (e.g., from demo 2) or the disabling of KASLR by specifying nokaslr in the Linux kernel command line options. |
| 6 | Demo 4 | This demo reads memory from another process by directly reading physical memory. For this demo, either we need the direct physical map offset (e.g., from demo 2), or we must disable KASLR by specifying nokaslr in the Linux kernel command line options. This program code can read arbitrary physical addresses. However, since physical memory contains a lot of non-human-readable data, a test tool (secret) is provided that puts a human-readable string into memory and directly provides the physical address of that string. |
| 7 | Demo 5 | This demo dumps the contents of memory (as in demo 3 and demo 4, this demo needs either the direct physical map offset from demo 2 or the disabling of KASLR by specifying nokaslr on the command line options of the Linux kernel) in a format similar to a hex dump. Again, since physical memory contains a lot of unreadable content, a test tool is provided to fill large amounts of physical memory with human-readable strings. |

For the experiment, a computer with an Elbrus-8S2 processor was used.

The binary translator of the Lintel system (TVGI.00509-01) was installed on the computer: a tool for launching operating systems in ×86 machine codes on computers with Elbrus architecture.

After that, Ubuntu operating system version 16.04 was downloaded from the official site and successfully installed.

As a result, the experimental stand had the following configuration:

- A personal computer with an Elbrus-8S2 processor;
- A binary translator of the Lintel system (TVGI.00509-01);
- An installed instance of the Ubuntu 16.04 operating system.

Demo No. 1 was performed. In the source, it is called "test". In a series of demos, the Meltdown vulnerability is the most basic. This demo exploits the Meltdown vulnerability to read available addresses from its own address space. In this case, no isolation mechanisms are violated. The authors point out that if this demo does not work, then the rest of the demos will most likely not work either [23].

Among the reasons why demos do not work is that the central processing unit may be too slow, may be an older generation and may not support the out-of-order execution of commands. It may be that the high-resolution timer is not accurate enough (especially in virtual machines). The operating system may not support custom signal handlers, among many other reasons.

For the demo, its code was compiled and run. For this, the following commands were executed:

*taskset 0 × 1./test.*

The source states that if, upon execution, output similar to this appears on the screen, then the basic demo works:

*Expect: Welcome to the wonderful world of microarchitectural attacks.*

*Got: Welcome to the wonderful world of microarchitectural attacks.*

On the Elbrus-8S2 processor, the basic Meltdown vulnerability demo 1 worked, and the reading of available addresses from its own address space was successful.

Next, we performed demo 2 by hacking Kernel Address Space Layout Randomization (KASLR). KASLR is the randomization of the location of the address space of the kernel image of the operating system at its boot time. Starting with kernel version 4.12 of the Linux operating system, this setting is active by default. This means that the location of the kernel (as well as the direct physical map showing all physical memory) in RAM changes every time the operating system is rebooted.

Demo 2 uses the Meltdown vulnerability to show the possibility of a leak (secret) when randomizing the location of the kernel address space. This demo requires root administrator privileges to speed up the process. The source provides an option that does not require raising the process level to the administrator (root rights).

For the demo, its code was compiled and run. For this, the following commands were executed:

*sudo taskset 0 × 1./kaslr.*

If the demo code is executed successfully, then after a few seconds, something similar to this should appear on the screen:

*[+] Direct physical map offset: 0xffff880000000000.*

On a computer with an Elbrus-8S2 processor and Lintel binary translator running Ubuntu version 16.04, demo 2 of the Meltdown vulnerability did not work. It was not possible to determine the direct physical map offset address.

Demos 3–5 also failed to produce results in the form of the possibility of exploiting the Meltdown vulnerability.

## 4. Discussion

As is known, the Meltdown vulnerability exploits a hardware error in the implementation of speculative command execution on some Intel processors. The vulnerability allows for ignoring access rights to memory pages during the speculative execution of commands and gaining unauthorized access to privileged memory, including that used by the operating system kernel.

The experiment showed that due to the peculiarities of the microarchitecture of Elbrus processors, the exploitation of Meltdown-type vulnerabilities is impossible. Moreover, unauthorized access to privileged memory is completely prevented. Unlike Intel processors, Elbrus microprocessors do not execute commands out of turn without checking access rights. Speculative execution of program code, according to the developers [28], is present, but it is implemented differently than in Western manufacturers. The specifics of the speculative and predictive modes of command execution, asynchronous access to arrays and other defining properties of the Elbrus microprocessor architecture, which allow high energy efficiency and performance to be achieved when setting the explicit parallelism of operations [28], occur synchronously with the installation of the required rights and privileges, which ensures high-level information security for data.

The key property of the Elbrus architecture related to the subject of the study can be considered the absence of the possibility of implicitly obtaining the contents of a memory cell (memory area) when accessing an uninitialized (invalid) memory area. In this case, a diagnostic value is generated, including in the event of a violation of the rules for restricting access to data.

It is possible to change the topology during production. In order to successfully control the implementation after receiving batches of microcircuits, the following actions during the development stages are advisable:

- Filling the free space left on the chip after the completion of the physical design process with active elements;
- Structuring the placement in the form of macroblocks in such a way as to be able to control the links and boundaries between macroblocks by existing controls;
- Transitioning to the use of CDB interfaces at the physical level and individual controllers designed by Russian enterprises.

When adapting free software for use as part of the Elbrus platform, taking into account the bypass of hardware vulnerabilities, it is an effective practice to apply general technical requirements to general software and general-purpose software. In particular, these requirements include the norms of GOST R 56939-2016 [29] when applied together with GOST R 58412-2019 [30]. In this standard, the following key norms can be distinguished:

- Requirements for the content and procedure for performing work related to the creation of secure software;
- Application of measures for the development of secure software throughout the entire life cycle (there is a connection to the processes described in GOST R ISO/IEC 12207-2010 [31]);
- Introduction of a basic set of measures for the development of secure software.

There are six types of software testing: static analysis and code review, functional program testing, introduction testing, dynamic code analysis and fuzzing testing.

The method for the elimination of vulnerabilities in computer facilities using the Lintel binary translator makes it possible to effectively neutralize a number of Meltdown-type vulnerabilities on the Elbrus microprocessor architecture, which was confirmed during the experiment.

## 5. Conclusions

The creation of control systems or automated systems for various purposes, as a part of the implementation of the import substitution policy in the Russian Federation and taking into account the pressure of sanctions from foreign states, takes place on the basis of Russian processors, common software and general system software. The basis for creating computer technology, for example, is the Russian hardware and software platform Elbrus. It is necessary to organize the management of the life cycle of vulnerabilities in all life cycle stages of processors and computer equipment. Vulnerability accounting solutions do not require the introduction of new stages, models or modeling tools but should become an integral part of development processes. The approach proposed in this study will solve the problem of developing new, "secure" processors in which the requirements for ensuring information security are implemented at the development stage.

Due to the peculiarities of the microarchitecture of Elbrus processors, the exploitation of Meltdown vulnerabilities is impossible. Moreover, unauthorized access to privileged memory is completely prevented. Solutions based on Russian microprocessors are undoubtedly more secure than those based on imported chips.

The results of this study show the effectiveness of the new method, which consists of using the architectural features of the Russian Elbrus computing platform and binary translation technology to eliminate the vulnerabilities inherent in the $\times 86$ architecture of computing technology from Intel.

Based on the new method, it is possible to develop a new algorithm and methods, the essence of which will be to search for and count the vulnerabilities found, for example, on a computer with an Intel processor and running a Windows operating system, with an attempt to further recreate this list of vulnerabilities on the same version of the Windows operating system but launched on a machine with Elbrus architecture using the Lintel binary translator. If the number of vulnerabilities decreases, then this means that it is the

vulnerabilities of the $\times86$ architecture (for example, computer technology from Intel) that have been found.

Modern computer technology is characterized by a constant reduction in the cost of preparing for production and the cost of mass production with an increase in the level of unification of computing modules, in which auxiliary electronic components and conditions for the operation of the processor are determined. For the effective control and accounting of vulnerabilities, it is important to increase the level of unification of modules, which is associated with a reduction in the range and economic feasibility of meeting the set of requirements for using the module as part of the hardware of various computer systems.

The creation of computing systems based on such processors largely determines their competitiveness, making it possible to reduce the design time.

The contribution of this study is that it complements the numerous studies by scientists around the world in the field of information security and cybersecurity. The existence of Elbrus hardware and software platforms as an alternative to large and widespread global brands, such as Intel, AMD and Advanced RISC Machine (ARM), will allow the creation of high-performance solutions that are trusted and protected from most malware (viruses, trojans, etc.).

**Author Contributions:** Conceptualization, I.B. and I.M.; methodology, P.K.; software, V.P. and A.R.; validation, I.B., I.M., P.K., V.P. and A.R.; formal analysis, A.R.; investigation, P.K.; resources, I.B.; data curation, I.M. and P.K.; writing—original draft preparation, I.B.; writing—review and editing, P.K. and V.P.; visualization, I.M.; supervision, A.R.; project administration, A.R.; funding acquisition, I.B., I.M., P.K., V.P. and A.R. All authors have read and agreed to the published version of the manuscript.

**Funding:** This research received no external funding.

**Data Availability Statement:** Not applicable.

**Conflicts of Interest:** The authors declare no conflict of interest.

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
