# Peer review of "Management of Hardware Vulnerabilities in the Life Cycle Stages of Microprocessors and Computers"

_inventions, doi:10.3390/inventions8040098_

Round 1

Reviewer 1 Report

In this paper, the authors discuss topical issues of managing information security vulnerabilities at the life cycle stages of processors and computer equipment. A list of points that appears to deserve to be better clarified in the paper together with some suggestions follows.

§  The manuscript is not clearly written and it is not well-organized. The authors should improve the structure of their paper.

§  The authors should describe in more detail and in clearly way the features of the development of processors, general software and general system software presented in section 2.

§  In the conclusions section the authors summarize the main points of their study. The authors should explain the contribution of their study.

§  The reference list is too short. There is a rich body of literature on all of the topics covered in this paper and many of these papers must be reported in the Introduction section.

§  The authors should refer to recent papers, such as the followings:

  • Gustavo Wolfmann, "Parallel Execution on Heterogeneous Multiprocessors from Algorithm Models Based on Petri Nets," Equations, vol. 1, pp. 47-54, 2021.
  • Atanas N. Kostadinov, Guennadi A. Kouzaev, "A Novel Processor for Artificial Intelligence Acceleration," WSEAS Transactions on Circuits and Systems, vol. 21, pp. 125-141, 2022.
  • Sujata. A.A, Lalitha. Y.S, "Design and Performance Analysis of 4-bit Nano-processor Design for Low Area, Low Power and Minimum Delay Using 32nm FinFET Technology," WSEAS Transactions on Electronics, vol. 12, pp. 1-8, 2021.

Author Response

1

1) The manuscript is not clearly written and it is not well-organized. The authors should improve the structure of their paper.

Answer: The article has the following structure:

1) There is the description of the problem of accounting for vulnerabilities in the created equipment. Examples of popular vulnerabilities in Intel processors and justification of the inconsistency of the fuzzing testing method application are given.

2) The results of the analysis of the features of processors development, general software and general system software, taking into account vulnerabilities, are presented. The process of verification of processors during development is considered.

3) The method for eliminating vulnerabilities using the architectural features of the Russian Elbrus computing platform (using binary translation technology) is formulated. A research hypothesis has been formulated, indicating that it is possible to confirm the claimed method using an experiment with an attempt to recreate Meltdown (rogue data cache load) CVE-2017-5754 vulnerability on the Elbrus hardware and software platform.

4) The detailed description of the experiment is given. The experiment showed that due to the peculiarities of the microarchitecture of Elbrus processors, the exploitation of Meltdown-type vulnerabilities is impossible. Moreover, unauthorized access to privileged memory is completely excluded.

5) The results of the analysis of the conducted experiment are presented. There are the conclusions on the work, which consist in the fact that the presented method of eliminating vulnerabilities in computer equipment using Lintel binary translator makes it possible to effectively neutralize a number of Meltdown-type vulnerabilities on Elbrus microprocessor architecture, which was confirmed during the experiment.

Based on the foregoing, additional changes to the structure of the manuscript, increasing its clarity of organization, are not necessary. The improvement of article structure, in our opinion, does not make sense.

2) The authors should describe in more detail and in clearly way the features of the development of processors, general software and general system software presented in section 2.

Answer: The materials describe the features of the development of processors, general software and general system software, taking into account the main topic of the article: accounting for vulnerabilities at the stages of the life cycle of processors and computer equipment. The article is related to the topic of information security, information protection, cybersecurity, but not to the topic of microprocessor development.

This means that additional materials are not required.

3) In the conclusions section the authors summarize the main points of their study. The authors should explain the contribution of their study.

Answer: The following part is added: the contribution of this study is that it complements the numerous studies of scientists around the world in the field of information security and cybersecurity. The existence of an alternative to such large and widespread global brands as Intel, Advanced Micro Devices (AMD), Advanced RISC Machine (ARM), Elbrus hardware and software platform will allow creating high-performance solutions that are trusted and protected from most malware (viruses, trojans, etc.).

4)  The reference list is too short. There is a rich body of literature on all of the topics covered in this paper and many of these papers must be reported in the Introduction section.

The following part is added: in the current topic of cybersecurity, which includes such areas as the detection of new threats to information security, vulnerabilities in systems and programs, attacks, intrusion detection in systems and software, network traffic anomalies and development of secure software, Russian and foreign authors, such as O.V. Kazarin, V.V. Lipaev, A.V. Barabanov, A.S. Markov, A.I. Kachalin, N.A. Gorbunov, V.L. Tsirlov, I.V. Kotenko, A.V. Lukatsky, I.V. Sharabyrov, E.S. Abramov, D.A. Andreev, M.L. Evgenievich, D. Yu. Gamayunov, A.N. Nazarov, Kh.A. Foelevich, A.A. Vladimirov, V.A. Galatenko, S.V. Gordeychik, S.A. Ermakova, P.D. Zegzhda, Constantinos Kolias, Muhamad E.A., Kwangjo Kim, M. Usha, etc. An analysis of the sources shows that the well-known works do not consider the features of the development of processors, common software and general system software, as well as taking into account vulnerabilities at the stages of the life cycle of processors and computer equipment. Not fundamental research, but developments aimed at practical application will increase the level of information security of various organizations and enterprises”.

The sources are also added to the references, the new ones are in color in the article.

5) The authors should refer to recent papers, such as the followings:

Gustavo Wolfmann, "Parallel Execution on Heterogeneous Multiprocessors from Algorithm Models Based on Petri Nets," Equations, vol. 1, pp. 47-54, 2021.

Atanas N. Kostadinov, Guennadi A. Kouzaev, "A Novel Processor for Artificial Intelligence Acceleration," WSEAS Transactions on Circuits and Systems, vol. 21, pp. 125-141, 2022.

Sujata. A.A, Lalitha. Y.S, "Design and Performance Analysis of 4-bit Nano-processor Design for Low Area, Low Power and Minimum Delay Using 32nm FinFET Technology," WSEAS Transactions on Electronics, vol. 12, pp. 1-8, 2021.

Answer: The article describes the features of the development of processors, general software and general system software, taking into account the main topic of the article: accounting for vulnerabilities at the stages of the life cycle of processors and computer equipment.

This article is related to the topic of information security and cybersecurity, but not to topics such as «Parallel Execution on Heterogeneous Multiprocessors from Algorithm Models Based on Petri Nets», «A Novel Processor for Artificial Intelligence Acceleration» or «Design and Performance Analysis of 4-bit Nano-processor Design for Low Area, Low Power and Minimum Delay Using 32nm FinFET Technology».

In our opinion this means that the reference of the above articles as additional sources is not advisable, but we added them to the list.

Reviewer 2 Report

This paper outlines the developed actions for controlling vulnerabilities and highlights the experiment's results, showcasing the effectiveness of the Elbrus processor's microarchitecture in mitigating Meltdown-type vulnerabilities.

However, the paper is extremely difficult to understand, so the authors need to work on the presentation with more effort.

More specifically, Fig 5 to Fig. 10, which are supposed to be results that demonstrate the claims by the authors, however carry no useful information at all to the reviewer.

Please improve the presentation. The current version is difficult to understand.

Author Response

1) The paper is extremely difficult to understand, so the authors need to work on the presentation with more effort.

Answer: The article has the following structure:

1) There is the description of the problem of accounting for vulnerabilities in the created equipment. Examples of popular vulnerabilities in Intel processors and justification of the inconsistency of the fuzzing testing method application are given.

2) The results of the analysis of the features of processors development, general software and general system software, taking into account vulnerabilities, are presented. The process of verification of processors during development is considered.

3) The method for eliminating vulnerabilities using the architectural features of the Russian Elbrus computing platform (using binary translation technology) is formulated. A research hypothesis has been formulated, indicating that it is possible to confirm the claimed method using an experiment with an attempt to recreate Meltdown (rogue data cache load) CVE-2017-5754 vulnerability on the Elbrus hardware and software platform.

4) The detailed description of the experiment is given. The experiment showed that due to the peculiarities of the microarchitecture of Elbrus processors, the exploitation of Meltdown-type vulnerabilities is impossible. Moreover, unauthorized access to privileged memory is completely excluded.

5) The results of the analysis of the conducted experiment are presented. There are the conclusions on the work, which consist in the fact that the presented method of eliminating vulnerabilities in computer equipment using Lintel binary translator makes it possible to effectively neutralize a number of Meltdown-type vulnerabilities on Elbrus microprocessor architecture, which was confirmed during the experiment.

Based on the foregoing, additional changes to the structure of the manuscript, increasing its clarity of organization, are not necessary. The improvement of article structure, in our opinion, does not make sense.

2) More specifically, Fig 5 to Fig. 10, which are supposed to be results that demonstrate the claims by the authors, however carry no useful information at all to the reviewer.

Answer: We thought a lot about the expediency of the figures presented in the article (from 5 to 10). Moreover, your question regarding these graphic materials is very clear to us.

During the internal discussion of the authors of the manuscript, it was decided that these materials should be left in the article.

Due to the fact that the article will be publicly available for reading and downloading on the Internet, it can be assumed that completely different people will read it. Perhaps there will be enthusiasts (young programmers, researchers of the possibilities of computer technology) who want to recreate the results we have obtained or even refute them. That is why the presence in the manuscript of "live" screenshots that prove and show our results is extremely necessary. We ask you to allow them to be included in the materials.

Round 2

Reviewer 1 Report

The paper can be published in present form.

Author Response

Dear reviewer,

Thank you very much.

Best regards,

Authors

Reviewer 2 Report

Thanks to the authors for the revision. However, this revision does not show too much improvement except for including more references. The major concerns, such as contribution and novelty, remain. Specifically, these figures are not that meaningful. If these figures are to show interested readers a way to reproduce, this paper should be publicized as a blog instead of a research article.

The introduction, method, and discussion are fine, but the authors need to think again about how to present the results appropriately. In conclusion, the reviewer will not recommend the current version for publication, even considering the quality of this journal. 

Again please do another proofreading for this paper, even using the software to improve the readability. 

Author Response

1) Thanks to the authors for the revision. However, this revision does not show too much improvement except for including more references. The major concerns, such as contribution and novelty, remain.

Answer: The text of the study has been supplemented.

The materials reflecting contributions and novelty are also added.

2) Specifically, these figures are not that meaningful. If these figures are to show interested readers a way to reproduce, this paper should be publicized as a blog instead of a research article.

Answer: The pictures you specified have been removed.

3) The introduction, method, and discussion are fine, but the authors need to think again about how to present the results appropriately.

Answer: The materials reflecting the obtained additional results are added.